# Canonical and Non-Canonical Roles of PFKFB3 in Brain Tumors

**DOI:** 10.3390/cells10112913

**Published:** 2021-10-27

**Authors:** Reinier Alvarez, Debjani Mandal, Prashant Chittiboina

**Affiliations:** 1Department of Neurological Surgery, University of Colorado School of Medicine, Aurora, CO 80045, USA; reinier.alvarez@cuanschutz.edu; 2Neurosurgery Unit for Pituitary and Inheritable Disorders, National Institute of Neurological Disorders and Stroke, Bethesda, MD 20824, USA; debjani.mandal@nih.gov; 3Surgical Neurology Branch, National Institute of Neurological Disorders and Stroke, Bethesda, MD 20824, USA

**Keywords:** PFKFB3, PFK-2, tumorigenic reprogramming, metabolic reprogramming, hypoxia, glycolysis, brain tumors, CNS

## Abstract

PFKFB3 is a bifunctional enzyme that modulates and maintains the intracellular concentrations of fructose-2,6-bisphosphate (F2,6-P2), essentially controlling the rate of glycolysis. PFKFB3 is a known activator of glycolytic rewiring in neoplastic cells, including central nervous system (CNS) neoplastic cells. The pathologic regulation of PFKFB3 is invoked via various microenvironmental stimuli and oncogenic signals. Hypoxia is a primary inducer of PFKFB3 transcription via HIF-1alpha. In addition, translational modifications of PFKFB3 are driven by various intracellular signaling pathways that allow PFKFB3 to respond to varying stimuli. PFKFB3 synthesizes F2,6P2 through the phosphorylation of F6P with a donated PO4 group from ATP and has the highest kinase activity of all PFKFB isoenzymes. The intracellular concentration of F2,6P2 in cancers is maintained primarily by PFKFB3 allowing cancer cells to evade glycolytic suppression. PFKFB3 is a primary enzyme responsible for glycolytic tumor metabolic reprogramming. PFKFB3 protein levels are significantly higher in high-grade glioma than in non-pathologic brain tissue or lower grade gliomas, but without relative upregulation of transcript levels. High PFKFB3 expression is linked to poor survival in brain tumors. Solitary or concomitant PFKFB3 inhibition has additionally shown great potential in restoring chemosensitivity and radiosensitivity in treatment-resistant brain tumors. An improved understanding of canonical and non-canonical functions of PFKFB3 could allow for the development of effective combinatorial targeted therapies for brain tumors.

## 1. Introduction

Profound molecular alterations associated with oncogenic transformation affect the metabolic and phenotypic profile, cell cycle, and the histological organization of human tumors [1]. In addition, tumor microenvironment propagates the oncogenic effects with phenotypic reprogramming of the surrounding milieu. The malformed tumoral vasculature, hypoxia, and starvation due to tumor growth may propagate tumorigenic reprogramming. Cancer cells exhibit dysregulated cell cycle mechanisms that allow for the evasion of regulatory checkpoint steps. Tumor metabolic reprogramming (TMR) with increased glycolytic flux is an important cancer hallmark, as it allows the cells to meet their increased energy demands while allowing for potential therapeutic targets. Proliferating tumor cells harmonize the process of metabolism and cell cycle progression, as glycolytic enzymes have all been shown to portray extra-metabolic abilities that facilitate cancer cellular proliferation and metastasis [2,3].

## 2. Hypoxic Reprogramming

Hypoxia-induced metabolic reprogramming, cell cycle progression, and angiogenesis are essential steps in tumorigenesis [4]. Tumor cells, particularly those in solid tumors, live in a spectrum of hypoxic conditions within their surrounding microenvironment. Tumors often outgrow innate vascular supply, and tumor-induced pathologic angiogenesis forms poorly structured vessels, together leading to a poor supply of oxygen and nutrients [5]. To overcome the limited oxygen availability, hypoxia-inducible factor (HIF) proteins are activated and promote downstream transcriptional events to adapt to the hypoxic environment with a preferential shift to aerobic glycolysis [6,7]. HIF-1α induces various glycolytic enzymes, including lactate dehydrogenase (LDH-A), hexokinase I and II (HK-I and HK-II), and glucose transporters I and III (GLUT-1 and 3), and modulates the regulatory bifunctional enzymes 6-phospho-2-kinase/fructose2,6-biphosphatase 1 through 4 (PFKFB1-4), with PFKFB3 being an essential modulator of tumorigenesis [8,9,10]. 

Glioblastoma neurospheres exposed to a hypoxic environment show a 10-fold increased expression of PFKFB3, as well as upregulation of LDHA, pyruvate dehydrogenase kinase 1 (PDK1), and HK-2, via HIF-1α signaling, particularly within the necrotic core [11,12]. Simultaneously, to maintain ATP levels during hypoxic conditions, glioblastoma cells upregulate PDK1 to reduce flux through the Krebs’s cycle and electron transport chain (ETC), thus favoring glycolysis [13,14]. HIF-1α-stimulated expression of monocarboxylate transporter 4 (MCT4) allows elevated intracellular lactate to be excreted, thereby circumventing intracellular acidification that could inhibit glycolytic enzymes while simultaneously supplying lactate as an energy substrate for nearby cells (termed “symbiotic fuel exchange”) [15]. 

The microenvironment within a solid tumor’s core region drastically differs from that of the peripheral margin [16] just as the metabolic phenotype of a particular tumor may vary from patient to patient [17]. The varying degree of hypoxia within a tumor makes intra-tumoral metabolic heterogeneity inevitable with a wide-spectrum of metabolic phenotypes [18,19]. Coincidentally, tumor reprogramming-induced glycolytic enzymes are ubiquitously expressed in healthy tissues, creating a therapeutic dilemma for selective targeting of tumor metabolism. Nonetheless, the nutrient-deficient tumor microenvironment and subsequent adaptations are key aspects to consider in tumorigenesis, cellular adaptation, and therapeutic targets.

## 3. PFKFB3

Metabolic reprogramming to maintain a constantly high rate of glycolytic flux is essential to tumorigenesis. The glycolytic rate-limiting enzyme 6-phosphofructo-1-kinase (PFK-1) is the first driver to commit a cell to glycolysis while being allosterically activated by fructose-2,6-bisphosphate (F2,6-P2) [20,21]. PFKFB3 is a bifunctional enzyme that essentially controls the rate of glycolysis by maintaining intracellular F2,6-P2 [21,22]. PFKFB3 has been shown to be one of the key factors involved in the glycolytic rewiring found in most cancer cells, including those in the central nervous system (CNS) [23]. In addition to TMR, PFKFB3 has been shown to be implicated in various oncogenic processes, such as tumorigenic angiogenesis [24], cell cycle progression [25,26] and DNA repair [26].

PFKFB3 is a member of the mammalian PFK-2/FBPase (PFKFB) family of enzymes made up by four isoenzymes (PFKFB1-4) [22,27,28]. PFKFB3 is considered essential for embryonal maturation and is expressed in cortical neurons, epithelial cells, and secretory cells in mouse embryos [29]. The PFKFB family members share an 85% sequence homology, but with differences in their tissue expression profiles, kinase/phosphatase activities, and overall response to molecular stimuli [30,31]. In adult mammals, PFKFB1 is predominately expressed in the liver and muscle; PFKFB2 in the heart; and PFKFB3 in the brain, placenta, and proliferating cells. By contrast, PFKFB4 is mostly expressed in the testis. However, tissue specificity is not absolute, as heterogenic expression is common. PFKFB is a family of homodimers containing two functionally different domains, the kinase domain (PFK-2) located in the N-terminal and the bisphosphatase domain (FBPase-2) located in the C-terminal. PFK-2 is the catalytic domain responsible for the synthesis of F2,6P2 while FBPase-2 performs the hydrolytic degradation of F2,6P2, both being essential steps in glycolytic regulation. The highly divergent functional domains provide much of the differences seen between each isoenzyme functionality. 

PFKFB3 has the highest kinase–phosphatase activity (710–740:1) of all PFKFB isoenzymes and is thus notorious in sustaining high glycolytic rates through the maintenance of elevated F2,6P2 levels [32]. PFKFB3 is ubiquitously expressed in human tissues, but with the highest concentration found in fat, muscle, the kidney, the lungs, and the brain [28,33]. It is encoded by the *PFKFB3* gene (NCBI Gene ID: 5209) located on chromosome 10p15.1 and contains 19 exons [34]. Multiple AUUUA instability elements are located on 3′UTR, allowing for facilitated post-transcriptional regulation [35]. As a consequence of alternative splicing at the variable C-terminal domain, six variants (UBI2K1-6) have been identified in humans [28,30]. Two of the splice variants show tissue preference: UBI2K3 is brain specific, while UBI2K4 is predominately found in skeletal muscle, although not exclusively [28]. A variety of transcription factor binding sites are located in the PFKFB3 promoter region, including E-box [36], early growth response 1(EGR-1) [12], specific protein 1 (Sp-1) [37], activating protein 2 (AP-2) [38], hypoxia response element (HRE) [39], ER response element (EREs) [40], progesterone response element [41], serum response element (SRE) [41], Krüppel-like factor 2 (KLF2) [42], cAMP response element-binding 1 (CREB1) [43], and E2F on a conserved F-type promoter [44]. Human vein endothelial cells also showed that PFKFB3 contains an antioxidant response element (ARE) transcriptional enhancer [45]. 

## 4. PFKFB3 Regulatory Mechanisms 

### 4.1. Transcriptional Regulation

Physiologic *PFKFB3* induction has been reported with adipocyte cell differentiation [46]. Pathologic regulation of *PFKFB3* is invoked via various microenvironmental stimuli and oncogenic signals. Of these, hypoxia is one of the primary inducers of PFKFB3 transcription with HIF-1α being the involved transcription factor binding at the HRE [8,12,47,48]. Deficient HIF-1α embryonic mouse fibroblast cells showed no PFKFB3 mRNA induction in a hypoxic environment as opposed to wild-type cells [8] Oncogenic signaling via the Ras pathway is a known regulator of cancerous glucose metabolism as well as a key contributor to the HIF-1α hypoxic response [9,49,50]. Ras signaling inhibition using Ras antagonist trans-farnesylthiosalicylic acid in murine glioblastoma cells showed downregulation of HIF-1α and *PFKFB3* with subsequent loss of glycolysis and diminished cell viability [9] The *PFKFB3* promoter also contains consensus response elements for the transcriptional factors estradiol [40] and progestin [41]. ETS domain transcription factor *PU.1* is a critical *PFKFB3* regulatory factor in tyrosine kinase inhibitor (TKI) resistance [51] E2F1 is a known transcriptional inducer of PFKFB3 by binding to the F-type promoter during G1/S phase progression [52]. PFKFB3 transcription has also been found to be rhythmically associated with the circadian cycle, as CLOCK transcription factor binds to the *PFKFB3* E-box promoter [36] Diminished tongue cancer growth in vivo was evident with PFKFB3 inhibition only at certain circadian time points [36]. As a product of HIF-1α signaling, high-grade gliomas (HGGs) have been shown to overexpress CLOCK, which seems to be associated with HGG TMR and radiotherapeutic resistance [53].

Constitutive activation of HER2 is a transcriptional driver of PFKFB3 in HER2+ breast cancer, with higher PFKFB3 mRNA being associated with a poorer progression-free survival [54]. Prolonged erlotinib treatment of human lung cancer cells showed increased PFKFB3 transcription by CREB1 recruitment mediated by MAPK signaling [43]. Insulin functions as a potent transcriptional inducer of PFKFB3 likely by binding to the various sterol regulatory sequences (SRE) and E-boxes in the promoter region of *PFKFB3* [55]. Inflammatory mediators, such as interleukin 6 (IL-6) through STAT3 signaling [56] and lipopolysaccharide (LPS) by binding to HRE, Sp-1, C/EBP, and NKκB domain s [57], are common microenvironmental stimuli shown to increase PFKFB3 transcription. Additional pro-inflammatory signals such as adenosine [57], mitogenic leptin (concanavalin A) [58], phytohemagglutinin [59], and TGF-β1 [60] also induce PFKFB3 transcription. Exposure to certain cellulotoxins (NaCL, UV radiation, and H_2_O_2_) produces a rapid and robust increase in PFKFB3 transcript by SRF, indicating PFKFB3 transcription as a potential cellular stress response [61]. PFKFB3 transcriptional up-regulation can also occur by yes-associated protein (YAP) acting as a co-activator in association with transcriptional enhancer activator domain 1 (TEAD1) in human endothelial cells [62].

Suppression of both PFKFB3 and PFKFB4 by induction of the gene expression regulator, myeloid translocation gene 16 (MTG16), led to glycolytic inhibition, the activation of mitochondrial respiration, and overall inhibition of cellular proliferation in B-lymphoblastoid Raji cells [63]. Wild-type p53 represses *PFKFB3* transcription as well as overall PFKFB3 protein expression [64,65]. PFKFB3 post-transcriptional regulation by miR-206 and miR-26b has been shown to reduce glycolysis, cell proliferation, and migration in cancer cells by direct interaction with the 3′UTR of PFKFB3 mRNA [66,67]. Inhibition of the liver X receptor (LXR) with a selective antagonist has been shown to also reduce both PFKFB3 mRNA and protein levels in vitro; however, the opposite when using an LXR agonist has not been described and, thus, requires further investigation [68].

### 4.2. Translational Regulation

Translational modifications of PFKFB3 are driven by various intracellular signaling pathways that allow PFKFB3 to respond to varying stimuli. Phosphorylation of Ser^461^ is essential for PFKFB3 kinase activity and is primarily initiated via AMP-activated protein kinase (AMPK) [48], protein kinase A (PKA) and C (PKC) [69], and mitogen-activated protein kinase-activated protein kinase 2 (MK2) [61]. Constitutively active and ligand-stimulated EGFRs increase PFKFB3 phosphorylation at Ser^461^ while also increasing overall PFKFB3 translation. Her2 activation has also been shown to increase PFKFB3 expression by Ras- and Akt-mediated signaling [54]. VEGF is also a known inducer of PFKFB3 protein expression, particularly in endothelial tip cells [24]. Under hypoxic conditions, phosphorylation at the Ser^461^ residue on the COOH terminal occurs through the AMPK pathway, while the p38/MK2 pathway is responsible for phosphorylation during stressful stimuli, such as NaCl, H_2_O_2_, and UV radiation [61]. PFKFB3 phosphorylation can, however, vary depending on nutrient availability. c-Src-mediated phosphorylation of PFKFB3 occurs at Tyr^194^ during periods of high glucose availability, while AMPK phosphorylation predominates in nutrient starved conditions [70]. AMPK phosphorylation also occurs during mitophagy-dependent removal of mitochondria and nutrient deficiency during prolonged mitotic arrest. Cyclin-dependent kinase (CDK) 6 is also involved in PFKFB3 phosphorylation, but at the Thr^463^ and Ser^467^ residue sites in recurrent breast cancer [71]. AMPK signaling has also been shown to induce PFKFB3 translation, particularly during mitosis through the activation of the PFKFB3 mRNA polyadenylation element in the 3′UTR, cytoplasmic polyadenylation element (CPE) [72].

PFKFB3 phosphorylation by inhibitor of nuclear factor kappa-B kinase subunit beta (IKKβ) has been shown to occur at Ser^269^ and inhibits PFKFB3 kinase activity. IKKβ-driven Ser^269^ phosphorylation decreases aerobic glycolysis during times of glutamine starvation and serves as a survival adaptation to minimize stress in times of nutrient scarcity [73]. The Cys^206^ residue located at the entrance of the PFKFB3 kinase catalytic pocket is subject to S-glutathionylation when in the presence of reactive oxygen species (ROS), leading to reduced catalytic activity for the production of F-2,6-BP [74]. S-glutathionylation of PFKFB3 at Cys^206^ causes a switch from cellular glycolysis to the pentose phosphate pathway (PPP). The preferential shift to the PPP has also been noted to occur in the presence of carbon monoxide (CO), as it prevents protein arginine methyltransferase 1 (PRMT1)-mediated PFKFB3 methylation at Arg^131^ and Arg^134^, causing inactivation of PFKFB3’s glycolytic control [75]. Di-methylated Arg^131^/Arg^134^ is known to facilitate glycolysis by enhancing F-6-P binding to Arg^133^ on the PFK-2 domain with subsequent production of F-2,6-BP [76]. In addition to glycolytic potentiation, di-methylated Arg^131^/Arg^134^ stabilizes PFKFB3 by preventing the polyubiquitination of Lys^142^ and, thus, prevents the eventual proteasomal degradation [77].

Post-translational degradation of PFKFB3 is orchestrated by two E3 ubiquitin ligases that recognize the KEN and DSG box consensus motifs [78]. The anaphase-promoting complex/cyclosome-cadherin 1 (APC/C-Cdh1) recognizes the KEN box, while SKP1-CUL1-F-box-protein (SCF) binds to the DSG box during the G1 and S phases of the cell cycle [78,79]. The phosphatase and tensin homolog (PTEN) tumor suppressor has been identified to facilitate PFKFB3 degradation by enhancing binding of Cdh1 to PFKFB3 [77]. Lack of a PTEN tumor suppressor function prevents cells from degrading PFKFB3 by the APC/C-Cdh1 mechanism. Long noncoding RNA (lncRNA) actin gamma 1 pseudogene (*AGPG)* stabilizes PKFB3 by binding to it and preventing APC/C-mediated ubiquitination, thereby preventing PFKFB3 proteasomal degradation [80]. 

### 4.3. PFKFB3 in Mammalian Tumorigenesis

PFKFB3 is highly expressed in many solid tumors and leukemia cells as well as in non-pathologic proliferating cells, albeit at a lower level [81]. Tumorigenesis associated with increased PFKFB3 transcription or with enhanced PFKFB3 phosphorylation has been shown to be implicated in ER+, PR+, HER2+, and recurrent breast cancer cells [40,41,54,66,72]. Oncogenicity of H-ras^V12^ has been found to be dependent on PFKFB3 expression [49]. Hypoxia-induced transcription of PFKFB3 as an oncogenic regulator is associated with increased proliferation and overall survival of pancreatic and gastric cancer cells [47]. PFKFB3 is not only an essential regulator of cell proliferation but also of increased cellular migration, noted in colorectal carcinoma cell lines through insulin signaling or IL-6 activation [55,82,83]. Aggressive invasiveness of pancreatic cancer cells directed by increased PFKFB3 also occurs as a result of TGFβ1 signaling [84]. Myelogenous cancer cells have been shown to be under constant growth stimulus through overly active PFKFB3 transcription, likely driven by JAK2/STAT5 phosphorylation in JAK2V617F kinase mutants [85]. Cancer stem cells (CSCs) can be distinguished from induced pluripotent stem cells (iPS) purely by the expression levels of PFKFB3 and PFK-1, as PFKFB3 overexpression is typical of CSCs [86]. Overall, cancer cells can adapt to stressful conditions and continue to proliferate in part due to the diverse and reversible functions of PFKFB3 (Figure 1). 

The involvement of PFKFB3 in cancer cell tumorigenic reprogramming is witnessed in various steps of cellular transformation, including metabolic reprogramming, cell cycle regulation, angiogenesis, and DNA repair. Recently, PFKFB3 has also been identified as a potential regulator in autophagy; however, whether it is a positive or negative regulator remains to be clarified [87,88,89].

### 4.4. PFKFB3 Metabolic Reprogramming: Glycolysis and Pentose Phosphate Pathway

The “Warburg effect” noted that neoplastic cells consumed large amounts of glucose and produced large amounts of lactate even in the presence of abundant oxygen, commonly referred to as aerobic glycolysis [90,91]. TMR with a preference for glycolysis is likely a response to the hypoxic environment that tumors experience with HIF-1α being a key pleiotropic inducer [92]. The overall rate of glycolysis is controlled by various molecular enzymes at different steps in the cycle of glucose utilization. The first step toward committed glycolysis is the phosphorylation of fructose-6-phosphate (F6P) to fructose-1,6-bisphosphate (F1,6P2) by 6-phosphofructo-1-kinase (PFK-1). Elevated intracellular levels of ATP or citric acid can inhibit PFK-1; however, F2,6P2, an allosteric activator, can reverse the inactivation of PFK-1 [89,93]. Not only does F2,6P2 prevent the inhibition of PFK-1 in the setting of high intracellular ATP, but it simultaneously increases the affinity for F6P. Constant intracellular availability of F2,6P2 can essentially override the cellular mechanisms in place to regulate glycolytic flux. 

The intracellular concentration of F2,6P2 in cancers is maintained primarily by the PFKFB3 bifunctional enzyme. PFKFB3 synthesizes F2,6P2 through the phosphorylation of F6P with a donated PO_4_ group from ATP and has the highest kinase activity of all PFKFB isoenzymes. PFKFB3 is overexpressed in a variety of aggressive human cancers and is one of the primary enzymes responsible for the glycolytic TMR [81]. Glycolysis provides tumor cells with various benefits, including rapid production of ATP, production of biosynthesis intermediates for de novo nucleic acid synthesis, and amino acid metabolism, while maintaining an acidic microenvironment [92]. Moreover, the acidic microenvironment produced by extracellular lactate concentrations degrades the adjacent extracellular matrix, promoting the invasion of surrounding structures and distant metastasis [94]. Tumorigenic acidosis can further promote mutagenic and clastogenic [95] behavior primarily via the inhibition of DNA repair [92]. Glycolytic TMR has been identified as one of the primary culprits in high-grade glioma (HGG) chemoresistance [96,97] and radioresistance [53,98,99]. Furthermore, TMR with a glycolytic preference instills a “domino effect” by mediating the aggregation of various glycolytic and chemoresistance activators (VEGF, IL-1β, IL-8, IL-6, TGF-β, and lactate) within the tumor stroma [100,101]. Cancerous cells employ glycolysis as a metabolic adaptation to meet the constant demand for cellular building blocks required for continuous and uncontrolled proliferation. 

PFKFB3 is also involved in the pentose phosphate pathway (PPP) as S-glutathionylation at Cys^206^ shunts glucose toward the PPP via the inhibition of the PFK-2 catalytic domain and diminished glycolytic flux [74]. It has been shown that the inhibition of PFKFB3 can also transition cells toward using the PPP, providing a backdoor mechanism for continued tumorigenesis through DNA repair [64]. Importantly, metabolic shift to the PPP serves as a survival adaptation to maintain cellular redox homeostasis. The PPP produces antioxidant-reduced glutathione (GSH) and nicotinamide adenine dinucleotide phosphate (NAPDH) through the oxidative arm, while the reductive arm produces nucleotides [102]. Elevated levels of ROS are a liability for tumor cells, as they induce cell cycle arrest [103] and autophagy [104]. The production of NADPH and GSH allows tumor cells to maintain redox homeostasis through ROS neutralization and allows for the detoxification of cancer cells. As a secondary option for cancer cells, PFKFB3 regulation in the setting of elevated ROS levels provides cells with the ability to escape the consequences of oxidative toxicity while continuing to use glucose. DNA repair is an additional benefit for the PPP employing cells, as it produces an abundance of nucleotides evidenced by increased amount of the precursor ribose-5-phosphate [64]. Nucleotide abundance allows for efficient DNA repair and increased survival following insults such as UV-mediated DNA damage. The seamless transition from glycolysis to the PPP offers tumor cells an overall selective advantage for continued proliferation [105].

Overall, PFKFB3 provides tumor cells with a reversible alternative that can provide protection against self-destruction and ensure cell survival simply by switching from biosynthesis to cellular redox.

## 5. Non-Canonical Roles of PFKFB3

### 5.1. PFKFB3-Mediated Angiogenesis

Angiogenesis is a common adaptation seen in most solid tumors. Pathological angiogenesis is the development of new blood vessels from preexisting vessels as tumor endothelial cells (TECs) proliferate. Similar to tumor cells, TECs implement a glycolytic metabolism even in the presence of oxygen. Of all glycolytic enzymes, PFKFB3 is the most abundant in endothelial cells, including TECs [24,106]. When compared to normal endothelial cells (ECs), TECs show a 3-fold higher glycolytic flux with overexpression of PFKFB3 and GLUT1 [24,107,108]. Vascular endothelial growth factor (VEGF) is a major pro-angiogenic factor in tumors and is a known transcriptional inducer and a potent post- transcriptional regulator of PFKFB3 [106,109].

Angiogenic vessel sprouting is orchestrated by tip cells that are at the leading edge of the vessel with long filopodia and stalk cells that proliferate to extend the branch; both are EC subtypes [24]. PFKFB3 induces tip cell differentiation and thus facilitates vessel sprouting. Even in stalk cell phenotype-induced ECs, PFKFB3 overexpression abrogated Notch intracellular domain (NICD)-induced pro-stalk cell behavior [24]. PFKFB3 is essential in tip cell localization, positioning, and lamellipodia formation with subsequent vascular sprouting [24,108]. Within the lamellipodia, PFKFB3 was found to congregate with the actin filaments responsible for initiation and orientation of lamellipodia, F-actin bundles, likely to quench the need for continuous ATP [24]. In addition, in a previous study endothelial tube formation required PFKFB3, as PFKFB3-knockdown human umbilical vein endothelial cells (HUVEC) had >40% diminished tube formation but were rescued by lactate addition to the culture medium [106]. 

These tumorigenic vessels often have “leaky” walls as a result of fewer pericytes [110] and lower levels of VE-cadherin [111], facilitating metastasis, local invasion, and surrounding edema. Metastatic potential is facilitated by PFKFB3 as it activates pericytes. Activated pericytes have diminished adhesive potential, as N-cadherin expression is only up-regulated during quiescence [107]. Junctional protein VE-cadherin is essential for the formation of a tight vascular barrier, which is prevented by increased PFKFB3 mediated endocytosis within EC. Cancer cell intravasation/extravasation is also largely influenced by PFKFB3, as it increases the expression of tumor cell adhesion molecules, VCAM-1, ICAM-1, and E- selectin by NF-κB signaling [107]. The increased adhesive capabilities of tumor cells facilitates their ability to enter surrounding blood vessels and induce necrosis of otherwise healthy and normal ECs.

The irregular shape and tortuosity of tumor vessels as well as the overall impaired function of the TECs create an environment with overall poor perfusion and limited oxygen delivery [5]. Diminished perfusion serves as a tumorigenic protector as it inhibits the delivery of chemo- and immuno-therapeutic drugs while exacerbating the existing hostile milieu as described in high-grade gliomas (HGGs) [112]. In addition, the hypoxic environment diminishes the formation of oxygen radicals necessary for DNA damage following treatment with radiation or chemotherapy [5].

### 5.2. PFKFB3 in Cell Cycle Regulation 

PFKFB3 is also a driver of tumorigenesis beyond its functions in the cytoplasm, as it holds a critical nuclear role in regulating cell cycle progression. PFKFB3 variant 5 (UBI2K5) contains a nuclear localization signal (NLS), containing KKPR amino acid residues at positions 472–475 in the C-terminal domain [25]. The NLS is recognized by importin α5, which binds at Lys^472^ and transports PFKFB3 through the nuclear envelope pore complex [113]. It has been observed that ATM-dependent PFKFB3 acetylation at Lys^472^ by PCAF (p300/CBP-associated factor), and GCN5 (general control non-depressible 5) impairs the nuclear localization signal leading to cytoplasmic PFKFB3 sequestration [25,113]. Cytosolic PFKFB3 retention increases Ser^461^ phosphorylation, leading to enhanced glycolytic activity. PFKFB3 acetylation protects cells from cisplatin-induced DNA damage, as it promotes glycolysis and allows for apoptotic evasion [113].

Nuclear localization of PFKFB3 modulates the cell cycle through its effects on cyclin-dependent kinases, particularly CDK1, CDK4, cyclin D3, and Cdc25c [26,114]. Cell proliferation promoted by PFKFB3 is dependent on both its nuclear localization and kinase activity. Yalcin et al. showed that inactivated kinase mutants (Arg^75^/Ala^76^) and NLS-inhibited mutants (Lys^472^/Ala^473^) failed to stimulate cellular proliferation, confirming that nuclear delivery of both PFKFB3 and intranuclear F-2,6-BP is required for cell cycle progression [26]. PFKFB3 increases intranuclear F2,6P2, allowing for CDK1 activation and subsequent Cdk-mediated phosphorylation of the Cip/Kip protein p27, which is then ubiquitinated and undergoes proteasomal degradation [26]. A reduction in p27 protein levels allows for G1/S, S/G2, and G2/M cell cycle progression, as p27 is a multi-step cell cycle suppressor and apoptotic activator [115,116]. Overexpression of Cdc25 by PFKFB3 can then ensure maintenance of CDK1 activation, as Cdc25 triggers CDK1 dephosphorylation [26,117], while cyclin D3 upregulation activates CDK4 and 6 [118]. PFKFB3 Lys^147^ residue binds to CDK4, leading to the formation of a stabilized HSP90/Cdc37/CDK4 complex that foregoes proteasomal degradation [119]. Stabilized CDK4 allows for further G1/S phase progression of the cell cycle. Subcellular localization of PFKFB3 into the nucleus does not have any direct metabolic regulatory effects; however, overexpression of lncRNA LINC00538 (YIYA) can initiate CDK6-mediated phosphorylation of PFKFB3 at Thr^463^ or Ser^467^ and promote further glycolysis in an indirect manner [75].

PFKFB3-mediated cell cycle progression presents a parallel proliferative mechanism for neoplastic cells. Persistent glycolysis can lead to NAPDH deficiency and excessive production of ROS, which can subsequently halt cell cycle. Additionally, the limited glucose availability and overall poor perfusion within tumors provide a less-than-ideal environment for cellular growth. As such, subcellular PFKFB3 nuclear localization allows cancer cells to continue cellular growth and proliferation even in a nutrient-poor environment.

### 5.3. PFKFB3 in DNA Repair

An additional non-canonical role of PFKFB3 has been recently identified in the DNA damage response pathway (DDR) for DNA repair [26,114]. As high levels of DNA damage induced by anticancer treatment cause cell cycle arrest and cell death, it behooves tumors to adapt. Tumor cells have thus acquired the ability to repair such DNA damage to evade death even in the presence of chemotherapeutic agents or ionizing radiation (IR). Interestingly, PFKFB3 expression has been found to occur mostly within the nucleus in certain cancers, such as hepatocellular carcinoma (HCC) [114]. Shi et al [114] conducted transcriptome analysis of G2/M stunted cells after PFKFB3 silencing and found that the PI3K/Akt signaling pathway was the most enriched. Furthermore, PFKFB3 has been found to directly bind and phosphorylates Akt, an essential activator of the DDR [114,120]. ERCC1-XPF is downstream to Akt and an essential endonuclease for nucleotide excision repair (NER), double-strand DNA break repair, homologous recombination (HR), and interstrand crosslinking (ICL) repair that is overexpressed by PFKFB3-mediated Akt induction [114,121]. PFKFB3 silencing has been observed to reduce the expression of Akt, pAkt, and ERCC1 with subsequent accumulation of DNA damage, tumor cell death, and reduced tumor growth [114]. Akt activation is also known to induce resistance to anticancer therapy via activation of DDR in HGG and glioblastoma [120,122]. 

Homologous recombination (HR) repair of double-strand DNA breaks following radiation therapy is also regulated by nucleic PFKFB3. Following IR, there is an increase in nuclear colocalization of PFKFB3 and γH2AX as a function of the DDR, orchestrated by the MRN complex (MRE11–RAD50–NBS1), ATM, and MDC1 [26]. Interestingly, overall PFKFB3 protein levels are unchanged; hence, PFKFB3 colocalization is purely due to nuclear recruitment. Complete HR repair requires the nuclear localization of RPA32, BRCA1, and RAD51, which only occurs in the presence of nucleic PFKFB3 foci [26]. PFKB3 silencing prior to IR decreases HR repair by 60% with prolonged accumulation of γH2AX foci, indicating reduced DNA repair and stunted G2/M phase progression [26]. Furthermore, PFKFB3 inhibition using 3PO, which does not inhibit PFKFB3 kinase activity, does not appear to affect HR repair, indicating that PFKFB3-mediated HR repair requires PFKFB3 kinase activity. In addition, ribonucleotide reductase regulatory subunit M2 (RRM2) is an essential enzyme for localized dNTP production in cells, particularly for DNA repair following IR [123]. However, RRM2 nucleic localization requires co-recruitment with PFKFB3, as PFKFB3 inhibition prevents RRM2 nucleic recruitment [26]. RRM2 colocalization with PFKFB3, however, is not required in non-irradiated conditions, promoting the possibility that RRM2 DNA repair mediated by PFKFB3 is a mechanism specific to radiotherapy resistance. Consequently, lack of PFKFB3 expression results in halted replication forks that are resumed upon nucleoside supplementation [26]. 

As such, PFKFB3 is essential in the DDR pathway, a commonly over-efficient process in treatment-resistant cancers. It appears that the supply of sufficient nucleotides is orchestrated by PFKFB3 as well as the recruitment of essential repair factors. CNS tumors such as high-grade gliomas (HGGs) commonly develop treatment resistance and, thus, recur following treatment. Although there is a paucity of PFKFB3-mediated chemotherapeutic and radioresistance data in CNS tumors, further characterization of such effects might prove fruitful. 

## 6. PFKFB3 in Central Nervous System Tumors

Central nervous system (CNS) tumors have an annual incidence rate of 23.79 per 100,000 with an average annual mortality of 4.42 per 100,000 [124]. Gliomas are the most common primary intracranial malignant tumors, with glioblastoma patients having a median survival of 14.6 months even with surgery, chemotherapy, and postoperative radiation [125]. High-grade gliomas (HGGs)/astrocytoma rely heavily on glycolysis, much more than other CNS tumors, partially due to their hypoxic microenvironment [53]. HGGs contain abundant, although dysfunctional, vascularization with poor oxygen supply and subsequent hypoxia, which is further exacerbated by the uncontrolled cellular proliferation. HGGs are known to overexpress HIF-1α as a result of their low oxygen environment evident through increased expression of hypoxic biomarkers VEGF and CA IX [126]. As such, efficacious delivery of chemotherapy is difficult with chemoresistance developing in most malignant CNS tumor patients. High PFKFB3 LOH at 10p14-p15, as well as low UBI2K4 expression levels in glioblastoma, has been shown to portend a poor prognosis with an overall poor survival [127,128,129,130].

PFKFB3 protein levels are significantly higher in HGGs than in non-pathologic brain tissue or lower grade gliomas, but without a relative upregulation of transcript levels [23]. Overly expressed PFKFB3 in gliomas may be due to their cellular origin. Glial cells, particularly astrocytes, have low APC/C-Cdh1 activity and thus are not able to efficiently degrade PFKFB3 [79]. In contrast, neurons are known to have a low glycolytic rate and are unable to mount a glycolytic stress response as a result of continuous APC/C-Cdh1 PFKFB3 proteasomal degradation. Glioblastoma CSCs have been noted to exhibit diminished APC/C-Cdh1 activity as a result of hyperphosphorylated Cdh1 affecting complex formation [131,132]. Diminished APC/C-Cdh1 ligase function with an accumulation of downstream substrates promoted invasion, proliferation, and self-renewal in CSCs [131]. Zscharnack et al [127] showed that PFKFB3 variant 5 (UBI2K5) is the predominant splice variant expressed in normal brain tissue. However, the majority of HGGs express both variant 5 (UBI2K5) and variant 6 (UBI2K6) [127]. PFKFB3 variant 4 (UBI2K4) is commonly downregulated in HGGs compared to low-grade and non-neoplastic brain. Exogenous overexpression of UBI2K4 has been observed to decrease cell viability and growth in glioblastoma U87 cell lines without inducing a glycolytic effect, as there were no changes in lactate production [127]. Interestingly, high PFKFB4 expression is correlated with improved survival in glioblastoma and neuroblastoma patients [128,130].

The most common chromosomal abnormality in glioblastoma is loss of heterozygosity (LOH) on the long arm of chromosome 10 [133]. LOH on 10p14-p15 results in allelic deletion of the *PFKFB3* gene with a resultant decrease in transcript levels and protein expression in 55% of glioblastomas [129]. Glioblastomas with PFKFB3 LOH show reduced expression of the growth-inhibiting UBI2K4 variant, which may explain the downregulated expression and associated poor prognosis observed in some HGGs. LOH on 10p23-q24 has also been documented in glioblastomas causing loss of PTEN, resulting in reduced post-translational degradation of PFKFB3, and this may explain the overexpression seen even with chromosome 10 LOH [77,134].

PFKFB3 transcriptional induction by E2F1 during cell cycle progression links metabolism with a cellular proliferative state by PFKFB3-induced mTORC1 activation [135]. Dysregulated E2F transcription factors, particularly E2F1, is associated with up-regulated tumorigenic genes in many cancers including CNS malignancy [136]. Oncogenic signaling by mTORC1 has been identified in pediatric low-grade gliomas [137,138], primary central nervous system lymphoma [139], glioblastoma [140,141], subependymal giant cell astrocytoma [142], and in Lhermitte–Duclos disease [143]. PFKFB3 induces mTORC1 activity through expression of pS6K and increases mTORC1 lysosomal translocation independent of AMPK regulation [135]. Loss of function mutation of tuberous sclerosis complex 1 (*TSC1*) or 2 (*TSC2*) leads to accumulation of Rheb GTPase, which subsequently activates mTORC1 [144]. TSC LOH, as well as the resulting mTORC1 activation, has been extensively documented in astrocytoma, ependymoma, ganglioglioma, oligodendroglioma, and glioblastoma [145]. Activated mTORC1 promotes the expression of HIF-1α, which subsequently increases PFKFB3 transcription and protein expression [144]. As PFKFB3 can be reciprocally induced by activated mTORC1, PFKFB3 appears to be involved in a vicious cycle of continuous overexpression and oncogenicity in various CNS tumors. However, the direct link between TSC LOH and PFKFB3 has not yet been studied in CNS tumors. However, PFKFB3 inhibition has been shown to decrease tumor formation in Tsc1- and Tsc2-null tumors implanted in a mouse model as well as diminishing the cellular proliferation and colony formation in vitro [144]. 

PFKFB3 expression has been shown to be more sensitive to HIF-1α expression than PFKFB4, which may explain the higher PFKFB3 expression observed in HGGs. A low PFKFB3:PFKFB4 mRNA ratio (7.7:1) has been found to be a poor prognostic factor in patients with *IDH*-wildtype primary glioblastoma [128]. Those with low PFKFB3:PFKFB4 ratios had a 9 month overall survival compared to 14 months for those with ratios greater than 7.7:1. High PFKFB3:PFKFB4 expression is associated with a 2.56 hazard ratio for worse overall survival and diminished evidence-free survival [130]. However, neuroblastomas with both high PFKFB3 expression and high PFKFB4 co-expression showed a 1.69 hazard ratio, indicating that PFKFB4 may offset negative effects caused by PFKFB3 [130].

Oncogenic signaling by Ras is a well-known culprit in CNS tumors and a key regulator of neural transformation [146]. Ras tumorigenic signaling is known to induce HIF-1α transcription and prevent its proteasomal degradation while also being a driver of PFKFB3 expression. Among glycolytic enzymes, PFKFB3 was the enzyme with the highest expression in glioblastoma [9]. Ras inhibition by trans-farnesylthiosalicylic acid (FTS) in glioblastoma U87 cell lines leads to transcriptional repression of genes involved in glycolytic metabolism as well as cellular proliferation and cellular communication. Blum et al [9] showed that the blunted glycolysis observed was primarily through the downregulation of HIF-1α-regulated genes, PFKFB3, and GLUT1 expression. PFKFB3 had the most prominent decrease with threefold diminished transcript levels [9]. Moreover, Ras oncogenesis has widespread effects on tumorigenic reprogramming and should thus be further investigated in relation to PFKFB3.

Moreover, PFKFB3 induction by TGF-β1 has been shown to be a key regulator of glioma reprogramming and colony formation [60]. A hyperactive Smad pathway by TGF-β1 signaling is associated with a poor prognosis and aggressive HGG behavior [147]. PFKFB3 overexpression in glioblastoma by TGF-β1 is primarily through transcriptional upregulation mediated by the Smad pathway [60]. However, inhibition of TGF-β1 and Smad does not abolish PFKFB3 induction. Rodriguez-Garcia et al [60] analyzed TGF-β1 downstream mediators and found that PFKFB3 induction also required p38/MAPK and PI3K/Akt to induce glycolysis. Diminished F-2,6-P2 concentrations were found to reduce the oncogenic potential of glioblastoma cells. Overall, increased TGF-β1 expression in HGG leads to transcriptional induction of PFKFB3, resulting in increased glucose uptake, increased F-2,6-P2 production, increased lactate production, and increased glycolytic flux. 

Although the data linking PFKFB3 tumorigenesis to CNS tumors are sparse, there are extensive data documenting the oncogenicity of many PFKFB3 inducers to poor prognosis in CNS neoplastic disease (Figure 2). 

Among the many molecular inducers in HGG, constitutive EGFR [148] (p. 38), [149], c-Src [150], and the JAK/STAT pathway [151] have been identified as oncogenic mediators in gliomas. In addition, IL-6 overexpression is associated with poor prognosis and drug resistance development in medulloblastoma [152]. Constitutive EGFR activation has also been identified in benign brain tumors [148]. Despite the genomic and oncogenic signatures of CNS tumors being heterogenous, targeting common and critical metabolic pathways via PFKFB3 could synergize targeted treatments. 

## 7. PFKFB3 Inhibitors for Cancer

A multitude of compounds that inhibit PFKFB3 have been synthesized [153]. However, not all PFKFB3 inhibitors have robust enough data for in-human use, and, as such, we will discuss only those with the most promising therapeutic potential for CNS tumors. In 2008, Clem et al. [154] synthesized a small-molecule chalcone-derived PFKFB3 inhibitor, 3PO (3-(3-pyridinyl)-1-(4-pyridinyl)-2-propen-1-one), and observed that it blocked glucose uptake and prevented G2/M phase progression in Jurkat T cell leukemia cells. 3PO was found to simultaneously decrease intracellular concentrations of F2,6BP, lactate, ATP, NAD+, and NADH, favoring an overall inhibitory effect on glycolysis. As a competitive inhibitor, 3PO suppresses the basal catalytic activity of PFKFB3 and suppresses glycolytic flux by decreasing the intracellular concentration of F2,6P2. In several human cancer cells, such as glioblastoma, tongue carcinoma, gastric cancer, and head/neck squamous cell carcinoma, 3PO exhibits antitumor activity while simultaneously reducing glucose metabolism [36,60,155,156,157,158]. However, due to poor pharmacokinetics [159] and lack of PFKFB3 selectivity [160], 3PO has not been further investigated in clinical trials. 

Various promising 3PO derivatives with similar molecular function have since been synthesized. For example, 1-(4-pyridinyl)-3-(2-quinolinyl)-2-propen-1-one (PFK15) is a potent derivative of 3PO, with a quinoline ring encompassing the ADP/ATP binding site of PFKFB3. PFK15 displays improved pharmacokinetic properties and PFKFB3 selectivity with 100-fold greater potency than that of 3PO with no organ toxicity verified by histological analyses [159]. Suppressed growth of U-87 glioblastoma xenograft tumors in mice has been observed with PFK15, but with less growth inhibition than seen with temozolomide [159]. Currently, an NIH SBIR R43 grant (Application #8392203) has been awarded to Advanced Cancer Therapeutics to further characterize the use of PFK15 in glioblastoma preclinical models. 

PFK-158 (1-(4-pyridinyl)-3-[7-(trifluoromethyl)-2E-quinolinyl]-2-propen-1-one), a derivative of PFK15, has been developed with an even greater half-life and diminished clearance, allowing for a lower dosage [153]. The use of PFK158 has shown broad antitumor activity in multiple preclinical models by also inhibiting cellular proliferation, glycolytic flux, and lactate production while synergistically inducing apoptosis [40,161,162,163]. The addition of PFK158 to anti-estrogen therapy enhanced the response to therapy, suggesting that improved and durable response may occur in ER+ tumor patients [40]. PFK158 diminished tumor burden, tumor volume, and overall tumor growth in malignant pleural mesothelioma (MPM) mouse xenografts, with no effect on overall mouse weight [162]. Improved pharmacokinetics with minimal off-site effects allowed PFK158 to become the first in-human PFKFB3 inhibitor for patients with advanced-stage solid malignancies (NCT02044861). Furthermore, three cohorts of the dose escalation phase I trial (NCT02044861) have been completed with no dose-limiting effects or drug-related side effects associated with PFK158. However, the status of this trial initiated in 2014 is currently unknown. 

In 2018, a phenylsulfonamide salicylic acid-derived small-molecule inhibitor, KAN0438241, that showed selectivity only for the PFKFB3 kinase domain was identified [26]. An esterified derivative, KAN0438757, showed increased cellular permeability with suppression of F-2,6-P2 and cell viability in pancreatic, gastric, and colorectal cancer cell lines. Normal cells showed tolerability to KAN0438757 treatment at concentrations that produced radiosensitivity and cytotoxicity in cancer cells. PFKFB3 inhibition with KAN0438757 after ionizing radiation prevented the subcellular nuclear localization of PFKFB3, BRCA1, RPA, RAD51, and RRM2, all shown to be essential HR repair molecules. KAN0438757 blocked HR repair activity and prevented dNTPs from being incorporated during double-strand break DNA repair, evidenced by persistently high levels of γH2AX and delayed recovery from IR-induced cell cycle arrest [26]. In- vivo administration of KAN0438757 did not show any systemic toxicity and was well tolerated even at the highest dose of 50 mg/kg in immune competent mice [164]. 

## 8. PFKFB3 Inhibitors for Treatment-Resistant Tumors 

Concomitant PFKFB3 inhibition has additionally shown great potential in restoring chemosensitivity and radiosensitivity in treatment resistant tumors [26,51,113,162,163,164,165]. Platinum-based chemotherapy resistance, which commonly occurs particularly through RAD51-induced HR, was reversed with the addition of PFK158 to the carboplatin/cisplatin chemotherapeutic regimen [163]. Cisplatin-resistant cells are known to increase PFKFB3 acetylation at Lys^472^ with subsequent cytoplasmic sequestration and phosphorylation in the presence of cisplatin treatment to avoid drug-induced apoptosis. The addition of PFK15 to cisplatin therapy rescued in vitro chemosensitivity in osteosarcoma, lung carcinoma, colorectal carcinoma, and endometrial cancer [113,163]. Combined PFKFB3 inhibition with oxaliplatin therapy suppressed tumor cell autophagy by decreasing p-AMPKα levels, a common protective adaptation in drug resistance [166]. Combined PFK158 and paclitaxel treatment on recurrent ovarian cancer mouse models sensitized the chemoresistant tumors by targeting both glycolytic and lipogenic pathways [161]. In addition to the cytotoxic effects of PFKFB3 inhibition, treatment of endometrial cancer cells with combined PFK158 and carboplatin/cisplatin synergistically downregulated Akt/mTOR signaling, common overexpressed mediators in CNS tumors [163]. 

Overexpression of receptor tyrosine kinases (RTKs) occurs in many human cancers, including brain tumors, and, thus, serves as a therapeutic target. However, resistance to tyrosine kinase inhibitor (TKI) therapy commonly occurs. PFKFB3 has been identified as one of the propagators of TKI resistance in tumor cells. In one study, combined therapy with PFK15 rescued BCR-ABLTKI-resistant leukemia cells in vitro and in vivo, as evidenced by the inhibited growth and prolonged survival of tumor xenograft mice [51]. TKI-sensitive cells showed no difference to TKI alone or combination therapy. Prolonged non-small-cell lung cancer (NSCLC) exposure to EGFR TKIs drives the upregulation of PFKFB3 transcription by CREB1 induction as a response to MAPK pathway activation to ensure cell survival [43,167]. PFK158 treatment increases re-established EGFR-TKI sensitivity at clinically relevant doses by cytotoxicity in NSCLC cells [43]. Combined PFKFB3 and EGFR inhibition abolishes the glycolytic response seen following EGFR-TKI-only treatment and prevents colony formation [43]. Resistance to sorafenib, a multi-kinase inhibitor, can present a challenge in the treatment of patients with HCC as a result of overexpressed PFKFB3. However, the addition of aspirin therapy was observed to sensitize HCC to sorafenib in vivo primarily through HIF-1α modulation leading to diminished PFKFB3 transcription and decreased tumor cell glycolysis [165]. Interestingly, aspirin-alone treatment has been found to decrease PFKFB3 protein, although to a lesser extent than when combined with sorafenib. As a result, targeting PFKFB3 in combination with TKIs may have clinical utility in the management of TKI-resistant brain tumors, such as malignant gliomas [168]. 

Patients with radiotherapeutic-resistant cancers have been found to share upregulated PFKFB3 transcript levels [26]. PFKFB3 inhibition with KAN0438757 prior to IR has been observed to decrease HR repair by 60% and increase radiosensitivity sixfold [103]. Radiosensitization by PFKFB3 inhibition is mediated by the abolishment of DNA repair factor RRM2 nuclear recruitment, which is essential following IR-induced DNA damage. Additionally, KAN0438757 abrogates PFKFB3s kinase activity, resulting in the prevention of nucleotide biosynthesis and required glycolysis [26]. Efficient DNA repair following IR has also been noted to occur in malignant CNS tumors. Radioresistance in HGG have also been linked to various signaling pathways associated with PFKFB3 expression, such as EGFR [169] and PI3K/Akt [170], which are associated with the development of radioresistance in HGG. mTORC1 activation is a known radioresistance propagator in pediatric gliomas [171]; thus, the mitigation of mTORC1 by PFKFB3 inhibition serves as a useful radiosensitizing modality. Targeted therapy to inhibit the PFKFB3 function appears promising for the induction of radiosensitivity in resistant cancers, such as malignant CNS tumors.

## 9. PFKFB3 Inhibition for Tumor Vessel Normalization

Traditional use of antiangiogenic therapy (AAT), such as the VEGF inhibitor bevacizumab, can hinder chemotherapeutic delivery while exacerbating tumorigenic hypoxia, thus creating a further hostile microenvironment [172]. Resistance to AAT also occurs, primarily through metabolic reprogramming that favors sustained glycolysis in both tumor cells and TECs [173,174]. Glioblastomas are known to upregulate PFKFB3 expression following anti-VEGF treatment as a form of AAT resistance [97]. Low-dose 3PO in vivo has been observed to improve the delivery and efficacy of chemotherapy by its tumor vessel normalization (TVN) effects. Excessive blockade of PFKFB3 can, however, facilitate metastasis and cancer cell extravasation by damaging the vascular integrity [175]. 

Tumor vessels of PFKFB3-silenced mice have been observed to contain considerable CD31^+^ laminin^+^ staining in the basement membrane, indicating vessel maturation [107]. PFKFB3 inhibition using low doses of 3PO optimized the vascular integrity and mural coverage by upregulating N-cadherin and increasing pericyte adhesion [107]. Anti-PFKFB3 treatment increased vessel lumen size and the total perfusable area of the tumor without altering the overall vessel density. PFKFB3 inhibition with 3PO was also observed to decrease tumor metastasis. Cantelmo et al [107] showed that 3PO reduced the number of tumor cells adhering to, and migrating across, ECs, as well as reducing the expression of cancer cell adhesion molecules VCAM-1, E-selectin, and ICAM-1. Novel PFKFB3 inhibitors (PA-1 and PA-2) synthesized as phenoxindazole analogues have been observed to inhibit angiogenesis via the downregulation of endothelial activators, adhesion molecules, and matrix metalloproteases MMP-2 and MMP-9 [176]. As a result of vessel maturation and improved EC layering, perfusion was improved in PFKFB3-silenced mouse models noted by reduced hypoxic markers. Interestingly, low dose 3PO did not affect tumor cell growth or proliferation, proving that the therapeutic effect was selective for TECs [107]. 

Dual inhibition therapy using low-dose 3PO and bevacizumab has been observed to increase survival, inhibit tumor growth, and prolong TVN duration in glioblastoma orthotopic xenograft mouse models [97]. Zhang et al [97] conducted a pathway enrichment analysis, showing that dual inhibition therapy downregulated the pathways involved in EGFR TKI resistance, chemokine mediation, Ras, and Rap1 signaling in glioblastoma tumors [97]. Glycolytic inhibition with low-dose 3PO has been observed to augment the effects of sub-optimal VEGF inhibition in vivo, showing a synergistic and additive effect of combined therapy [108]. Significant reductions in lipid/creatine have been observed following dual inhibition therapy, suggesting reduced tumor cell proliferation and necrosis [97]. Tie1, a regulatory protein involved in VEGF-independent tumorigenic angiogenesis, has also found to be downregulated upon dual therapy in glioblastoma cells [97]. Dual-inhibitory therapy has also been demonstrated to alleviate tumor hypoxia and lactate production. Bevacizumab monotherapy, however, has been observed to promote a further hypoxic environment with a preference for glycolysis and elevated lactate production [174]. Interestingly, treatment with PA-1 and PA-2 inhibits VEGFA/VEGFR2 and may provide the function of dual-inhibition therapy with a single pharmaceutical; however, in-human studies are pending [176]. Additionally, dual inhibition therapy has been observed to promote normalization of the vascular function with diminished aberrant permeability, thus allowing for improved chemotherapeutic delivery and reoxygenation [97]. Reoxygenation is known to dampen metastatic potential, chemoresistance, radioresistance, and metabolic reprogramming [5]. Combining PFKFB3 inhibitors with anti-VEGF therapy have been noted to prolong the effects of TVN by threefold compared to bevacizumab monotherapy, providing an extended window of opportunity for chemotherapeutic delivery [97].

Overall, PFKFB3 inhibition induces EC glycolysis and structural adaptations that facilitate tumorigenic angiogenesis. Inhibition of PFKFB3 as a method for TVN is thus a promising anti-tumorigenic treatment, as TVN has been shown to increase survival in glioblastoma patients [112,177].

## 10. PFKFB3 Inhibition for Immune Modulation

Immunotherapy for various cancers has been shown to prolong survival, although therapeutic resistance can occur and lead to eventual disease progression [178,179]. Myeloid-derived suppressor cells (MDSCs) have been identified as key regulators of immunotherapy in malignant CNS tumors [180,181]. MDSCs suppress T cell activation and cytotoxicity while promoting the differentiation and expansion of regulatory T cells [182]. In addition to T cell control, MDSCs inhibit natural killer cell activation while promoting the differentiation of anti-inflammatory macrophage (M2) phenotypes [183]. Murine and human MDSCs have been found to contain high PFKFB3 expressions [184]. 

MDSC-mediated T cell immunosuppression occurs as a result of arginase 1 (ARG1) and inducible nitric oxide (iNOS) transcriptional upregulation, enzymes known to suppress T cell function [185]. The hypoxic tumor microenvironment found in most solid tumors, including malignant CNS tumors, drives the upregulation of HIF-1α, which is a known inducer of MDSC differentiation in addition to being an ARG1 and iNOS activator [185]. Additionally, tumor hypoxia with resulting acidosis is known to abrogate the function of surrounding immune cells via the induction of anti-immunogenic mediators (e.g., IL-1b, IL-8, IL-6, TNFα, and TGF-b) contributing to tumor cell escape of immunogenic control [186]. PFKFB3 inhibition with PFK158 has been observed to diminish the suppressive ability of CD14^+^ HLA-DR ^low/-^ M-MDSCs in patients with stage III/IV melanoma [184]. Unpublished data by Chesney et al [184] show that PFK158 diminished the immunosuppressive properties of MDSC by dampening the glycolytic metabolic reprogramming while also suppressing ARG1 levels [184]. It has recently been noted that PFKFB3 silencing abolished Arg1 production in M2 macrophages [187]. 

Malignant gliomas, such as glioblastomas, have shown to be a rather immune-resistant tumors through a variety of innate and acquired characteristics [188]. Elevated MDSCs in female glioblastoma patients have recently been found to be associated with a poor prognosis [189]. Although the data on PFKFB3 overexpression in MDSC are currently scarce, this line of study appears to be promising. Further investigation on possible PFKFB3 roles in CNS tumor immunotherapeutic resistance can provide an additional treatment modality to the neuro-oncology armamentarium. 

## 11. Conclusions

Recently, Faubert et al. [190] explained the importance of metabolic therapy in tumorigenesis. Almost a century ago, Otto Warburg and his colleagues noted increased glucose consumption and the production of lactate in tumor cells compared to non-proliferating normal cells. Since then, this aerobic glycolytic phenotype has been regarded as a cancer hallmark and is termed the “Warburg effect”. Most tumors have a hypoxic microenvironment because of poor perfusion from abnormally formed tumor blood vessels. When oxygen levels are depleted, HIF-1α regulates a shift from oxidative metabolism to aerobic glycolysis. This hypoxic-driven glycolytic preference further stimulates a tumorigenic microenvironment through cellular secretion of lactate. Elevated lactate in the surrounding environment diminishes chemotherapeutic delivery while stimulating tumorigenic angiogenesis via the activation of HIF-1α-mediated upregulation of VEGFR2 [191]. TMR is essential to tumorigenesis, as it allows for a constantly high rate of glycolytic flux to be maintained. 

HIF-1α activates a multitude of oncogenic signaling pathways, one of which is PFKFB3 induction. PFKFB3 has the highest kinase activity of all PFKFB isoenzymes and is thus able to sustain continuous glycolysis through elevated F2,6P2 levels; constant F2,6P2 levels override mechanisms that normally control glycolytic flux [34]. The intracellular concentration of F2,6P2 in cancers is primarily maintained by PFKFB3, allowing cancer cells to evade glycolytic suppression. In addition to providing continuous energy for uncontrolled proliferation, glycolytic TMR is responsible for chemoresistance [94,95] and radioresistance [55,96,97] in a variety of cancers, including malignant CNS tumors. In addition to glycolytic rewiring, PFKFB3 is implicated in angiogenesis [26], continuous cell cycle progression [27], and cancer cell DNA repair [28]. PFKFB3 overexpression has been associated with a poor prognosis in various neoplastic diseases.

In brain tumors, including high-grade gliomas, PFKFB3 expression has been linked to poor survival [127,128,129,130]. However, targeted PFKFB3 inhibition may provide a therapeutic option for CNS tumor patients. PFKFB3 inhibitor therapy has been observed to restore chemosensitivity and radiosensitivity in treatment-resistant tumors [26,51,113,162,163,164,165]. Combined anti-PFKFB3 and anti-VEGF therapy has been noted to improve the survival of glioblastoma preclinical models and abrogate resistance to anti-angiogenic therapy [97]. The use of PFKFB3 inhibitors has synergistic effects with chemotherapeutic and radiotherapeutic modalities via mitigation of TMR, promoting anti-tumor cytotoxicity, prolonging TVN, and modulating immune suppressive cells. An improved understanding of PFKFB3 metabolic and extra-metabolic functions and their relation to tumorigenicity may allow for the development of targeted therapy as a means of modulating its various oncogenic behaviors.

## Figures and Tables

**Figure 1 cells-10-02913-f001:**
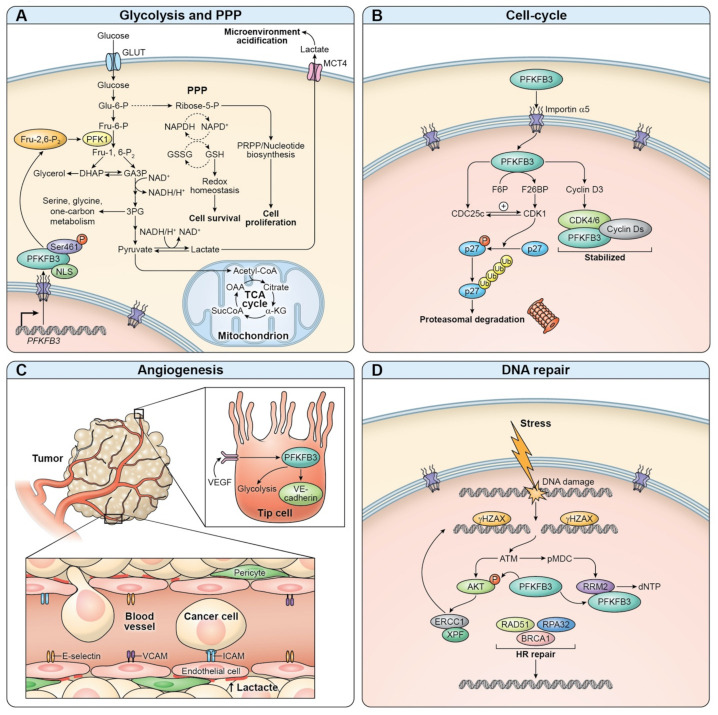
Metabolic and extra-metabolic functions of PFKFB3. The metabolic functions of PFKFB3 include driving aerobic glycolysis while also stimulating the pentose phosphate pathway (**A**). Extra-metabolic functions of PFKFB3 include cell cycle regulation (**B**), angiogenesis (**C**), and DNA repair after insult (**D**). Artist: Ethan Tyler, Medical Arts, Office of Research Services, National Institutes of Health, Bethesda, MD, USA.

**Figure 2 cells-10-02913-f002:**
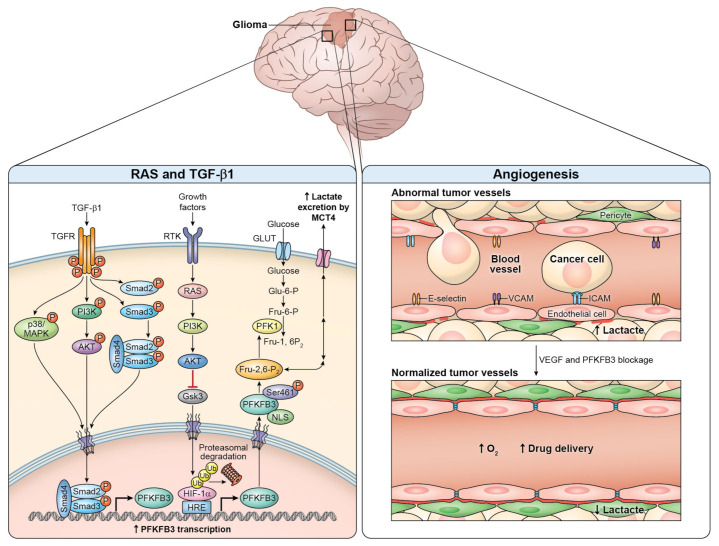
PFKFB3 tumorigenesis in gliomas. Oncogenic mediators RAS and TGF-β1 have been found to be primary drivers of PFKFB3 overexpression in high-grade gliomas. Tumorigenic angiogenesis in gliomas is promoted by PFKFB3 expression in tumor endothelial cells. Artist: Ethan Tyler, Medical Arts, Office of Research Services, National Institutes of Health, Bethesda, MD, USA.

## Data Availability

Not applicable.

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
