# Peer review of "Canonical and Non-Canonical Roles of PFKFB3 in Brain Tumors"

_cells, 2021, doi:10.3390/cells10112913_

Round 1

Reviewer 1 Report

In the manuscript entitled “Canonical and Non-Canonical Roles of PFKFB3 in CNS Tumors”, Alvarez et al. described the versatile roles of PFKFB3 in malignant brain tumors. This is a comprehensive review based on the available literature. Some comments below may help to improve the manuscript.

  • The body's CNS consists of the brain and the spinal cord. What are the roles of PFKFB3 in spinal cord tumors?
  • On page 13, the authors stated “Suppressed growth of U-87 glioblastoma xenograft tumors in mice was observed with PFK15, similar to temozolomide treatment and with the ability to cross the blood brain barrier.[166]”. However, Clem et al. concluded “…the activity of PFK15 against glioblastoma growth was lower than that observed by temozolomide (Fig. 5B)…” in [166]. This preclinical study may raise concerns how to improve the therapeutic potential of PFKFB3 inhibitors in the treatment of brain tumors.  
  • There are six ubiquitous PFKFB3 splice variants with variable nucleotide sequences and tissue distributions in humans. The expression level of individual PFKFB3 isoforms is not always correlated positively with brain tumor growth. Given the existence of such variant heterogeneity in malignant brain tumors, the therapeutic effectiveness of non-PFKFB3 variant-specific inhibitors may be limited.
  • Listed below are some minor errors:

Abstract: Fructose-2, 6-bisphosphate (F2,6-P2) is misplaced.

Figure 1 legend: “…..cell- cycle regulation (B),angiogenesis (C), ….”.

Figure 2 legend:  “……PFKFB3 overexpression in high-grade gliomas Tumorigenic angiogenesis……”.

Author Response

In the manuscript entitled “Canonical and Non-Canonical Roles of PFKFB3 in CNS Tumors”, Alvarez et al. described the versatile roles of PFKFB3 in malignant brain tumors. This is a comprehensive review based on the available literature. Some comments below may help to improve the manuscript.

Thank you for taking the time to review our article and for your comments/suggestions. Please find them addressed below.

  • The body's CNS consists of the brain and the spinal cord. What are the roles of PFKFB3 in spinal cord tumors?

Response: PFKFB3 involvement in spinal cord tumors have not been documented, however, it has been studied in the setting of traumatic spinal cord injury. By inhibiting neuronal apoptosis and increasing glycolysis PFKFB3 expression has been shown to have a neuroprotective effect in traumatic spinal cord injury in rats. Reference provided below

Gao L, Wang C, Qin B, Li T, Xu W, Lenahan C, Ying G, Li J, Zhao T, Zhu Y, Chen G. 6-phosphofructo-2-kinase/fructose-2,6-bisphosphatase Suppresses Neuronal Apoptosis by Increasing Glycolysis and "cyclin-dependent kinase 1-Mediated Phosphorylation of p27 After Traumatic Spinal Cord Injury in Rats. Cell Transplant. 2020 Jan-Dec;29:963689720950226. doi: 10.1177/0963689720950226. PMID: 32841050; PMCID: PMC7563815.

  • On page 13, the authors stated “Suppressed growth of U-87 glioblastoma xenograft tumors in mice was observed with PFK15, similar to temozolomide treatment and with the ability to cross the blood brain barrier.[166]”. However, Clem et al. concluded “…the activity of PFK15 against glioblastoma growth was lower than that observed by temozolomide (Fig. 5B)…” in [166]. This preclinical study may raise concerns how to improve the therapeutic potential of PFKFB3 inhibitors in the treatment of brain tumors.
  • There are six ubiquitous PFKFB3 splice variants with variable nucleotide sequences and tissue distributions in humans. The expression level of individual PFKFB3 isoforms is not always correlated positively with brain tumor growth. Given the existence of such variant heterogeneity in malignant brain tumors, the therapeutic effectiveness of non-PFKFB3 variant-specific inhibitors may be limited.
  • Listed below are some minor errors:

Abstract: Fructose-2, 6-bisphosphate (F2,6-P2) is misplaced. 

Corrected to: PFKFB3 is a bifunctional enzyme that modulates and maintains the intracellular concentrations of Fructose-2, 6-bisphosphate (F2,6-P2), essentially controlling the rate of glycolysis.

Figure 1 legend: “…..cell- cycle regulation (B),angiogenesis (C), ….”. Corrected

Figure 2 legend: “……PFKFB3 overexpression in high-grade gliomas Tumorigenic angiogenesis……”. Corrected

Reviewer 2 Report

In this review article, Alvarez and colleagues review the role of PFKFB3 in “tumors” rather than focusing on CNS tumors. This review article includes much research related to PFKFB3 but is not well-organized, lacks clarity, uses inappropriate citation, and conveys wrong information to readers. On a general comment, it would be helpful if the authors had provided line numbers so that the review process could be expedited.

Major issues:

  1. I am not sure whether it is appropriate that copy some sentences directly from other parts (3. PFKFB3 “fructose-2, 6-bisphosphate (F2,6-P2). PFKFB3 is a bifunctional enzyme that modulates and maintains the intracellular concentrations of F2,6P2, essentially controlling the rate of glycolysis. PFKFB3 has been shown to be one of the key factors involved in the glycolytic rewiring found in most cancer cells, including those in the central nervous system (CNS).” into Abstract.
  2. Poor emphasis is given to the role of PFKFB3 in CNS tumors which is a generic term encompassing over 120 distinct tumor types.
  3. Some issues involved in CNS tumor progression should be discussed in more detail including PFKFB3 protein levels are significantly higher in high-grade glioma without a relative upregulation of transcript levels.
  4. Inappropriate citation including “Multiple AUUUA instability elements are located on the 3’UTR allowing for facilitated transcription. [36]”.
  5. Additionally, figure legends are brief, it is difficult to know and understand them.
  6. Wrong information including “Multiple AUUUA instability elements are located on the 3’UTR allowing for facilitated transcription. [36]” (corrected: through post-transcriptional regulation), “High PFKFB3 expression is linked to poor survival and increased malignancy in CNS tumors. [36,134–136]” (corrected: Ref.36, the PFKFB3 LOH as well as the resulting low UBI2K4 expression level was associated with a poor prognosis of glioblastoma patients.), “Suppressed growth of U-87 glioblastoma xenograft tumors in mice was observed with PFK15, similar to temozolomide treatment and with the ability to cross the blood-brain barrier. [166]” (corrected: the activity of PFK15 against glioblastoma growth was lower than that observed by temozolomide.), “Gliomas are the most common primary intracranial malignant tumors with glioblastoma patients having a median survival of 8 months regardless of treatment.” (corrected: 14.6 months)
  7. When the same name is used more than once in a paper, the abbreviation may be used in the second and subsequent uses of the name. The abbreviation should be consistent throughout.
  8. There would be a number of strange statements such as: “PFKFB3 has the highest kinase:phosphatase activity (710-740:1) of all PFKFB isoenzymes”, “PFKFB3 Metabolic Reprogramming: Glycolysis and Pentose Phosphate Pathway”, “In addition, endothelial tube formation requires PFKFB3 as PFKFB3KO had >40% diminished tube formation”, “Interestingly, high PFKFB4 expression is correlated with improved survival in glioblastoma and neuroblastoma patients.”

Author Response

Reviewer 2

In this review article, Alvarez and colleagues review the role of PFKFB3 in “tumors” rather than focusing on CNS tumors. This review article includes much research related to PFKFB3 but is not well-organized, lacks clarity, uses inappropriate citation, and conveys wrong information to readers. On a general comment, it would be helpful if the authors had provided line numbers so that the review process could be expedited.

Major issues:

  1. I am not sure whether it is appropriate that copy some sentences directly from other parts (3. PFKFB3 “fructose-2, 6-bisphosphate (F2,6-P2). PFKFB3 is a bifunctional enzyme that modulates and maintains the intracellular concentrations of F2,6P2, essentially controlling the rate of glycolysis. PFKFB3 has been shown to be one of the key factors involved in the glycolytic rewiring found in most cancer cells, including those in the central nervous system (CNS).” into Abstract.

Corrected See abstract as sentences were edited and removed accordingly from abstract or text.

  1. Poor emphasis is given to the role of PFKFB3 in CNS tumors which is a generic term encompassing over 120 distinct tumor types.

  1. Some issues involved in CNS tumor progression should be discussed in more detail including PFKFB3 protein levels are significantly higher in high-grade glioma without a relative upregulation of transcript levels.

  1. Inappropriate citation including “Multiple AUUUA instability elements are located on the 3’UTR allowing for facilitated transcription. [36]”.

Corrected citation:

Chesney, J.; Mitchell, R.; Benigni, F.; Bacher, M.; Spiegel, L.; Al-Abed, Y.; Han, J.H.; Metz, C.; Bucala, R. An inducible gene product for 6-phosphofructo-2-kinase with an AU-rich instability element: Role in tumor cell glycolysis and the Warburg effect. Proc. Natl. Acad. Sci. U. S. A. 1999, 96, 3047–3052, doi:10.1073/pnas.96.6.3047.

  1. Additionally, figure legends are brief, it is difficult to know and understand them.

  1. Wrong information including “Multiple AUUUA instability elements are located on the 3’UTR allowing for facilitated transcription. [36]” (corrected: through post-transcriptional regulation),

Corrected to: Multiple AUUUA instability elements are located on the 3’UTR allowing for facilitated post-transcriptional regulation.[36]

“High PFKFB3 expression is linked to poor survival and increased malignancy in CNS tumors. [36,134–136]” (corrected: Ref.36, the PFKFB3 LOH as well as the resulting low UBI2K4 expression level was associated with a poor prognosis of glioblastoma patients.)

Corrected to: High PFKFB3 LOH at 10p14-p15 as well as low UBI2K4 expression levels in glioblastoma, have been shown to portend a poor prognosis with an overall poor survival.[134–137]

“Suppressed growth of U-87 glioblastoma xenograft tumors in mice was observed with PFK15, similar to temozolomide treatment and with the ability to cross the blood-brain barrier. [166]” (corrected: the activity of PFK15 against glioblastoma growth was lower than that observed by temozolomide.),

Corrected to: Suppressed growth of U-87 glioblastoma xenograft tumors in mice was observed with PFK15, however, with less growth inhibition than seen with temozolomide.[167]

“Gliomas are the most common primary intracranial malignant tumors with glioblastoma patients having a median survival of 8 months regardless of treatment.” (corrected: 14.6 months)

Corrected to: Gliomas are the most common primary intracranial malignant tumors with glioblastoma patients having a median survival of 14.6 months even with surgery, chemotherapy, and postoperative radiation.[132]

  1. When the same name is used more than once in a paper, the abbreviation may be used in the second and subsequent uses of the name. The abbreviation should be consistent throughout.

  1. There would be a number of strange statements such as: “PFKFB3 has the highest kinase:phosphatase activity (710-740:1) of all PFKFB isoenzymes”, “PFKFB3 Metabolic Reprogramming: Glycolysis and Pentose Phosphate Pathway”, “In addition, endothelial tube formation requires PFKFB3 as PFKFB3KO had >40% diminished tube formation”, “Interestingly, high PFKFB4 expression is correlated with improved survival in glioblastoma and neuroblastoma patients.”

Reviewer 3 Report

This is a well written and very extensive review if the role of PFKB3 in CNS tumors. In addition to the prose, there are two high quality illustrations demonstrating key pathways related to the canonical and non-canonical roles of PFKB3. My concerns are mostly superficial and can be addressed in a minor revision: 

  • The abstract begins with a dangling “Fructose-2,6-bisphosphate (F,2,6-P2)”. This should likely be moved into the first sentence- the period is aberrant.
  • Introduction
    • The use of the term “subpar” referring to tumoral vasculature in the introduction could be clearer – this is better addressed in the following section in which the authors describe pathologic angiogenesis as forming “poorly structured” vessels, and they should consider applying such language to the earlier reference
  • Hypoxic Reprogramming
    • The first reference to PDK1 is not defined – it is defined in the following sentence
    • There should be parentheses around the first definition of the acronym “ETC"
  • PFKFB3 in Cancer
    • On page 7, it would be useful to briefly comment on the proposed mechanism by which PFKFB3 S-glutathionylation shunts glucose toward the PPP
    • On page 7, it is unclear what is meant by “switching from biosynthesis to cellular redox” as the function of PFKFB3 in cancer remains biosynthetic, albeit at higher levels and for different substrates
  • PFKFB3 Mediated Angiogenesis
    • This section delineates angiogenesis as a non-canonical role of the PFKFB3 enzyme, but then relates the mechanism of this angiogenic impact to its impact on glycolytic flux, itself a canonical function of the enzyme. This may be more appropriately presented as a consequence of the canonical PFKFB3 function. This dependence on the metabolic functions of PFKFB3 is demonstrated by a study later referenced, in which supplementation of lactate reversed the impact of PFKFB3 knockout on endothelial tube formation
  • PFKFB3 in Cell Cycle Regulation
    • Page 9 refers to “Overexpression of Cdc25 by PFKFB3”, but PFKFB3 is not a transcription factor involved in the direct upregulation of Cdc25. Additionally, reference 123 does not support this direct link between PFKFB3 and Cdc25 over expression.
    • On page 9, the use of the word “subpar” is non-specific. A better term may be “nutrient-poor"
  • PFKFB3 in DNA Repair
    • Experiments describing the necessity of PFKFB3 for PI3K-Akt induction and subsequent resistance to DNA damage should be described if published – silencing without rescue is not enough to exclude e.g. off-target activity of a silencing modality or a collateral consequence of cell cycle perturbation
    • Discussion of PFKFB3 recruitment to the site of HR uses the term “increased PFKFB3 expression” but subsequently states that protein levels are unchanged. A more appropriate term may be “colocalization"
  • PFKFB3 Inhibitors for Cancer
    • The authors comment on the blood brain barrier penetrance of PFK15, bu they should also comment on the blood-brain barrier penetrance of PFK-158 and KAN0438241
    • The authors should comment on presence or absence of observed toxicities of PFK15 in preclinical models
  • PFKFB3 Inhibitors for Treatment Resistant Tumors
    • The authors should briefly comment on the specificity of the effect of aspirin via the PFKFB3 axis for modulation of sorafenib sensitivity in HCC

Author Response

Reviewer 3

This is a well written and very extensive review if the role of PFKB3 in CNS tumors. In addition to the prose, there are two high quality illustrations demonstrating key pathways related to the canonical and non-canonical roles of PFKB3. My concerns are mostly superficial and can be addressed in a minor revision: 

  • The abstract begins with a dangling “Fructose-2,6-bisphosphate (F,2,6-P2)”. This should likely be moved into the first sentence- the period is aberrant.

Corrected

  • Introduction
    • The use of the term “subpar” referring to tumoral vasculature in the introduction could be clearer – this is better addressed in the following section in which the authors describe pathologic angiogenesis as forming “poorly structured” vessels, and they should consider applying such language to the earlier reference

Corrected to: The malformed tumoral vasculature, hypoxia, and starvation due to tumor growth may propagate tumorigenic reprogramming.

  • Hypoxic Reprogramming
    • The first reference to PDK1 is not defined – it is defined in the following sentence

Corrected to: Glioblastoma neurospheres exposed to a hypoxic environment show a 10-fold in-creased expression of PFKFB3, as well as upregulation of LDHA, pyruvate dehydrogenase kinase 1 (PDK1), and HK-2, by HIF-1a signaling, particularly within the necrotic core.[11,12]

    • There should be parentheses around the first definition of the acronym “ETC"

Corrected to: Simultaneously to maintain ATP levels during hypoxic conditions, glioblastoma cells upregulate PDK1 to reduce flux through the Krebs’s cycle and electron transport chain (ETC) thus favoring glycolysis.[13,14]

  • PFKFB3 in Cancer
    • On page 7, it would be useful to briefly comment on the proposed mechanism by which PFKFB3 S-glutathionylation shunts glucose toward the PPP

Corrected to: PFKFB3 is also involved in the pentose phosphate pathway (PPP) as S-glutathionylation at Cys206 shunts glucose towards the PPP through inhibition of the PFK-2 catalytic domain and diminished glycolytic flux.[106]

    • On page 7, it is unclear what is meant by “switching from biosynthesis to cellular redox” as the function of PFKFB3 in cancer remains biosynthetic, albeit at higher levels and for different substrates

Response: PFKFB3 does maintain a relatively constant level of biosynthesis, however, when ROS levels are elevated causing PFKFB3 to switch towards PPP rather than glycolysis, production of antioxidants of GSH and NADPH are increased.

  • PFKFB3 Mediated Angiogenesis
    • This section delineates angiogenesis as a non-canonical role of the PFKFB3 enzyme, but then relates the mechanism of this angiogenic impact to its impact on glycolytic flux, itself a canonical function of the enzyme. This may be more appropriately presented as a consequence of the canonical PFKFB3 function. This dependence on the metabolic functions of PFKFB3 is demonstrated by a study later referenced, in which supplementation of lactate reversed the impact of PFKFB3 knockout on endothelial tube formation
  • PFKFB3 in Cell Cycle Regulation
    • Page 9 refers to “Overexpression of Cdc25 by PFKFB3”, but PFKFB3 is not a transcription factor involved in the direct upregulation of Cdc25. Additionally, reference 123 does not support this direct link between PFKFB3 and Cdc25 over expression.

Corrected to: Overexpression of Cdc25 by PFKFB3 can then ensure maintenance of CDK1 activation as Cdc25 triggers CDK1 dephosphorylation [26,123] while cyclin D3 upregulation activates CDK4 and 6[124].

    • On page 9, the use of the word “subpar” is non-specific. A better term may be “nutrient-poor"

Corrected: As such, subcellular PFKFB3 nuclear localization allows cancer cells to continue cellular growth and proliferation even in a nutrient -poor environment.

  • PFKFB3 in DNA Repair
    • Experiments describing the necessity of PFKFB3 for PI3K-Akt induction and subsequent resistance to DNA damage should be described if published – silencing without rescue is not enough to exclude e.g. off-target activity of a silencing modality or a collateral consequence of cell cycle perturbation
    • Discussion of PFKFB3 recruitment to the site of HR uses the term “increased PFKFB3 expression” but subsequently states that protein levels are unchanged. A more appropriate term may be “colocalization"

Corrected to: Interestingly, overall PFKFB3 protein levels are unchanged, hence PFKFB3 colocalization is purely due to nuclear recruitment.

  • PFKFB3 Inhibitors for Cancer
    • The authors comment on the blood brain barrier penetrance of PFK15, but they should also comment on the blood-brain barrier penetrance of PFK-158 and KAN0438241

Response: There are no studies currently describing the blood-brain barrier penetrance for these two drugs, however, PFK-158 is known to be more lipophilic than PFK15 and should theoretically exhibit similar if not greater penetrance. Further literature on the ability for KAN0438241 to penetrate the blood-brain barrier is not available.

    • The authors should comment on presence or absence of observed toxicities of PFK15 in preclinical models

Corrected to: PFK15 displays improved pharmacokinetic properties and PFKFB3 selectivity with 100-fold greater potency than 3PO with no organ toxicity verified by histological anal-yses.[167]

  • PFKFB3 Inhibitors for Treatment Resistant Tumors
    • The authors should briefly comment on the specificity of the effect of aspirin via the PFKFB3 axis for modulation of sorafenib sensitivity in HCC

Response: Authors did not emphasize this as it is out of the scope of this review

Round 2

Reviewer 1 Report

Brain tumors, not CNS tumors, should be used in the title, since PFKFB3 involvement in spinal cord tumors is not discussed.

What is your response to the comment below?

  • There are six ubiquitous PFKFB3 splice variants with variable nucleotide sequences and tissue distributions in humans. The expression level of individual PFKFB3 isoforms is not always correlated positively with brain tumor growth. Given the existence of such variant heterogeneity in malignant brain tumors, the therapeutic effectiveness of non-PFKFB3 variant-specific inhibitors may be limited.

Author Response

Brain tumors, not CNS tumors, should be used in the title, since PFKFB3 involvement in spinal cord tumors is not discussed.        

Response:

We agree with the reviewer’s astute observations. We have now edited the title and discussion to reflect our focuso n brain tumors rather than CNS tumors.

There are six ubiquitous PFKFB3 splice variants with variable nucleotide sequences and tissue distributions in humans. The expression level of individual PFKFB3 isoforms is not always correlated positively with brain tumor growth. Given the existence of such variant heterogeneity in malignant brain tumors, the therapeutic effectiveness of non-PFKFB3 variant-specific inhibitors may be limited.

Response:

This is precisely why we believe that this topic is of utmost importance. If we wish to provide new therapeutic options for patients with brain tumors, particularly, high grade gliomas, we need to have a deeper understanding on how to exploit PFKFB3 modulation for therapeutic potential. Our main objective with this review is to provide awareness and show that PFKFB3 modulation seems promising but further research is needed. We hope to motivate and engage the scientific community with pursuing dedicated research on PFKFB3.

Reviewer 2 Report

The authors have addressed the minor issues listed but did not respond to some questions mentioned in the first round of reviews.

  1. PFKFB3 may play an important role only in glioma but is not documented in other CNS tumors including spinal cord tumors or meningiomas. Some descriptions including the “Title” using the term “in CNS tumors” may not be appropriate.
  2. As mentioned in the Abstract that “PFKFB3 protein levels are significantly higher in high-grade glioma than in non-pathologic brain tissue or lower-grade gliomas, however, without a relative upregulation of transcript levels.” But there is still a focus on transcriptional regulation in Figure 2 (the graph is labeled Figure 4). I believe that an additional Figure(s) on post-transcriptional or post-translational regulation of PFKFB3 is necessary.
  3. Inappropriate placement and duplicate description issue: Page 10,

“High PFKFB3 LOH at 10p14-p15 as well as low UBI2K4 expression levels in glioblastoma, have been shown to portend a poor prognosis with an overall poor survival expression is linked to poor survival and increased malignancy in CNS tumors.[134–137]”

“Glioblastomas with PFKFB3 LOH show reduced expression of the growth-inhibiting UBI2K4 variant which may explain the downregulated expression and associated poor prognosis observed in some HGG.”

  1. Wrong information including“In addition, endothelial tube formation requires PFKFB3 as PFKFB3KO had >40% diminished tube formation” (corrected: in PFKFB3-knockdown HUVECs).

Author Response

The authors have addressed the minor issues listed but did not respond to some questions mentioned in the first round of reviews.

  1. PFKFB3 may play an important role only in glioma but is not documented in other CNS tumors including spinal cord tumors or meningiomas. Some descriptions including the “Title” using the term “in CNS tumors” may not be appropriate.

Response:

  1. As mentioned in the Abstract that “PFKFB3 protein levels are significantly higher in high-grade glioma than in non-pathologic brain tissue or lower-grade gliomas, however, without a relative upregulation of transcript levels.” But there is still a focus on transcriptional regulation in Figure 2 (the graph is labeled Figure 4). I believe that an additional Figure(s) on post-transcriptional or post-translational regulation of PFKFB3 is necessary.

Response: Literature on PFKFB3 is overall limited, as such we did not include a figure on this topic as the literature and data is sparse. In high grade gliomas, the literature available does not dive deeply into the steps or molecular processes accounting for these changes and as such we rather not depict steps/processes that may be inaccurate or without the full picture on such a topic.

  1. Inappropriate placement and duplicate description issue: Page 10,

“High PFKFB3 LOH at 10p14-p15 as well as low UBI2K4 expression levels in glioblastoma, have been shown to portend a poor prognosis with an overall poor survival expression is linked to poor survival and increased malignancy in CNS tumors.[134–137]”

“Glioblastomas with PFKFB3 LOH show reduced expression of the growth-inhibiting UBI2K4 variant which may explain the downregulated expression and associated poor prognosis observed in some HGG.”

Response: Authors do not believe that this is duplicate information as the former sentences is merely introducing the topic and the latter statement provides further detail on the causative effect.

  1. Wrong information including“In addition, endothelial tube formation requires PFKFB3 as PFKFB3KO had >40% diminished tube formation” (corrected: in PFKFB3-knockdown HUVECs).

Response: Thank you for the correction, it has been corrected in the text in the corresponding sentence.